# Neuropsychological Study in Patients with Spinal Cord Injuries

**DOI:** 10.3390/healthcare9030241

**Published:** 2021-02-24

**Authors:** Brígida Molina-Gallego, Sagrario Gómez-Cantarino, María Idoia Ugarte-Gurrutxaga, Laura Molina-Gallego, Laura Mordillo-Mateos

**Affiliations:** 1FENNSI Group, Hospital Nacional de Parapléjicos, SESCAM, 45071 Toledo, Spain; Laura.Mordillo@uclm.es; 2Nursing Department, Hospital Nacional de Parapléjicos, SESCAM, 45071 Toledo, Spain; 3Nursing Department, Faculty of Physiotherapy and Nursing, University of Castilla-La Mancha, 13071 Toledo, Spain; sagrario.gomez@uclm.es (S.G.-C.); Maria.ugarte@uclm.es (M.I.U.-G.); 4Nursing Department, Hospital Mancha-Centro, SESCAM, Alcázar de San Juan, 13071 Ciudad Real, Spain; laumoga82@gmail.com; 5Faculty of Health Sciences, University of Castilla-La Mancha, 13071 Talavera, Spain

**Keywords:** spinal cord injury, mild cognitive impairment, neuropsychological test

## Abstract

The present investigation was designed to determinate the nature, pattern, and extent of cognitive deficits in a group of participants with subacute and chronic spinal cord injury (SCI). Methods: A cross-sectional study was conducted in both patients with subacute and chronic SCI. Different cognitive functions were evaluated through a neuropsychological protocol designed for this purpose, taking into account the patient’s emotional state. Results: A total of 100 patients suffering a spinal cord injury were evaluated. There were no differences between the two groups when age, sex, level of education, and region of origin were studied. The chronic injured patients obtained lower scores in the neuropsychological evaluation protocol respective to the subacute injured patients. Conclusions: Subjects with chronic spinal cord injury presented a cognitive profile that differed greatly in the number of altered cognitive functions as well as in their magnitude from the subacute spinal cord injured patient profile. Moreover, cognitive dysfunction may be important beyond the end of the first stage of rehabilitation as it can affect an individual’s quality of life and possible integration in society.

## 1. Introduction

Spinal cord injury (SCI) occurs when the spinal cord is severely bruised, compressed, lacerated, or severed as a result of traumatic injury or disease. SCI is associated with the development of secondary conditions such as chronic pain, infections, and chronic fatigue, all of which contribute to lowered quality of life and potentially reduced social participation [1,2,3,4].

Between 40% and 60% of patients with subacute spinal cord injury (SCI) have demonstrated several types of cognitive deficits [5,6,7]. These deficits occur in areas of memory, capacity of attention, processing speed, and executive function and are independent of the level of SCI. Recent investigations in this issue show an association between cognitive deficits and adverse changes in the cardiovascular and cerebral systems in spinal-cord-injured patients [8]. Prospective and retrospective studies have examined or observed cognitive problems or dysfunctions in patients with spinal cord injuries from various perspectives [9,10] and suggest that these deficits affect in a negative way the rehabilitation program and the social integration of people with SCI [8].

However, there are few studies that clarify the nature or etiology of the deficits in individuals with SCI. Several factors can contribute to the occurrence of these deficits such as alcoholism, substance abuse, previous learning problems, the use of psychoactive medication, or emotional problems [6,11,12,13,14]. Although many of these are often transient, others like those related to problems in mental understanding and processing may persist for the first months after SCI [12].

Regarding emotional problems, different studies carried out showed that anxiety and depression levels in patients with spinal cord injury were between 19% and 30% in the first years after spinal cord injury and approximately between 23% and 38% suffer from depression. These data must be taken into account to rule out cognitive deficits associated with these states, and thus be properly diagnosed in order to predict the success of the rehabilitation treatment [14].

The recognition of the presence and nature of these deficits during the initial stage after SCI is important because it is in this period when the maximum rehabilitation stage takes place [15].

Rehabilitation of patients with SCI is an intense process that includes training in personal care, mobility, community skills, as well as physical and psychosocial teaching in adapting to disability. For this reason, any obstacle in learning and adaptation that induce problems or deficits at cognitive level, can compromise the achievement of optimal results in rehabilitation [15]. Previous studies stablished a relationship between cognitive impairment and traumatic brain injury in SCI. However, further studies have underlined the potential impact to other causes of cognitive impairment [8,16,17,18]. Regardless of the cause, the main result of these deficits in the brain is the fact that they are potentially harmful to the rehabilitation processes and that their effects are minor and remediable if they are recognized early.

Neuropsychological assessments can identify these problems early as well as their severity.

The term mild cognitive impairment (MCI) was introduced to define the clinical situation of cognitive decline that does not reach the intensity of dementia and is estimated not to be caused by aging but by an underlying pathology [19,20,21]. Therefore, it is characterized by a cognitive defect, which generally involves memory but is not severe enough to satisfy the necessary criteria for the diagnosis of dementia.

After the long-term follow-up of these patients, different types of MCI have been evidenced. Currently three subtypes are recognized:Amnestic MCI (aMCI), characterized by an isolated memory deficit.Multiple-domain MCI (aMCI-MD), which implies a slight deficit of more than one cognitive domain, may or may not include memory, but does not meet the criteria for the diagnosis of dementia.Single-domain MCI (aMCI-SD), which represents the effect of a single domain being different from memory [22].

The neuropsychological pattern of each MCI, understood as those affected domains, allows the sub-classification of the MCI into subtypes. Several groups have reported the magnitude and nature of the abnormalities observed in neuropsychological tests in SCI patients, using a wide range of assessments for cognitive function [6,11,12,23,24].

The present study was designed with the following objectives:To determine the nature, pattern, and extent of cognitive impairment in a group of subacute SCI patients and to compare it with another group of chronic SCI patients, using a comprehensive battery of reliable and validated neuropsychological assessment to study a broad range of cognitive functions. A neuropsychological assessment protocol designed for this purpose with validated tests was used, allowing evaluation of the different cognitive functions.To compare the cognitive deficits present in the subacute stage of SCI with those seen in the chronic stage, matched for age, sex, and educational level.

Our main hypothesis is that some cognitive deficits directly or indirectly caused by the SCI may improve with time, others can be stable over time, and others could worsen in the chronic stage. So, from a clinical point of view, it is important to detect the presence of cognitive disfunction as it may interfere with the first stage of the rehabilitation, which is the most intense and important one. Also, cognitive impairment can affect an individual´s quality of life and possible full integration to society, something of vital importance in the life of SCI patients.

## 2. Materials and Methods

A cross-sectional study was conducted.

### 2.1. Participants

One hundred SCI patients hospitalized at the Hospital Nacional de Parapléjicos in Toledo, Spain, were recruited from February 2012 to December 2013.

Fifty participants (30 male, age 46.82 ± 15.77) were in the subacute stage and fifty were in the chronic stage (31 male, age 47.80 ± 13.75).

In the subacute stage, we included participants with a recent first-time admission to our SCI unit with a time from injury ranging from four to six months. In the chronic stage, we included participants with a time from injury of at least one year, that were attending the hospital for normal annual follow-up.

The participants with SCI were recruited from the “National Hospital for Paraplegics”, an SCI rehabilitation hospital. The individuals with subacute stage were recruited when admitted to the hospital for rehabilitation. The group with chronic SCI was recruited from individuals who were attending the hospital for annual follow-up.

Inclusion criteria for both groups consisted of (1) patients with spinal cord injury of any etiology and of both sexes; (2) age at injury of 18 years or older; (3) age of 18–85 years at the time of interview; (4) Spanish speaking; (5) score at the mini-mental state exam (MMSE) of 22 or more (see neuropsychological assessment); (6) time since injury of more than four months for subacute SCI patients and at least one year for chronic ones; (7) injury level below C4, including ASIA C (motor function is preserved below neurologic level and more than half of the key muscle groups below neurologic level have a muscle grade less than 3) and ASIA D (motor function is preserved below neurologic level and at least half of the key muscle groups below neurologic level have a muscle grade of 3) [25].

Exclusion criteria included: (1) no radiological evidence of SCI; (2) injury level above C4 or depending on mechanical ventilation or diaphragmatic pacemaker; (3) critically ill patients; (4) age of injury younger than 18 years; (5) the presence of clinically demonstrated TBI ( Traumatic Brain Injury) (due to the known association of TBI with cognitive dysfunction); (6) severe psychiatric disorders; (7) history of central or peripheral neurological problems prior to the SCI; (8) known history of alcohol and drug abuse; (9) aphasia or other language disorders, as well as hearing and/or visual impairment; (10) time since injury inferior to four months for subacute SCI patients and twelve months for chronic ones; (11) score at the mini-mental state exam (MMSE) of under 22.

The study was approved by the local ethical committee and evaluated favorably by the Research Commission of the Hospital Nacional de Parapléjicos. Patients were informed about the study and all of them gave written, informed consent.

### 2.2. Research Protocol

Prior to initiating the study, the whole procedure was verbally explained to each participant emphasizing the motivation and objectives of the study. In addition, they were informed about the possibility of withdrawing from the study at any time, and data protection, as well as privacy, were guaranteed. An information sheet explaining the whole procedure was provided to the participants and all of them gave written, informed consent. When the participant was unable to sign, an oral consent was collected and a family member or legal guardian signed the written one.

All participants included in the study were assessed in the context of the same initial assessment protocol, which included:In a first stage: collection of information on sociodemographic aspects, such as family and personal history, toxic habits, factors related to the SCI, and its consequences. Information on aspects or factors related to hospitalization was also collected.In a second stage: neuropsychological assessment protocol was carried out. The order of administration of the tests, within each session, followed rules established by the characteristics of each one of them. The protocol was applied following an order in all cases. Average duration: 90 min.

This evaluation was performed when the patient was in an optimal condition—this date was estimated to be four months after admission, for the group of subacute SCI patients [26]. For the chronic group, the test was performed during their hospital stay for annual follow-up, in a scheduled appointment, when the participant did not suffer any complications.

### 2.3. Study Design

This is a cross-sectional study comparing participants with SCI in the subacute and chronic stages matches for age, sex, and educational level (case control study design).

Clinical and demographic data:Demographic information was collected from each participant including: age, sex, education (read/write, primary school, secondary school and university). Also, each participant was questioned regarding a history of high-frequency alcohol and/or substance use [27,28].Clinical information included: lesion level (cervical, dorsal, and lumbar), American Spinal Injury Association Impairment Scale (AIS) grading (A, B, C, D, E), and time since injury (months). Participants were also categorized as to whether or not they were taking at least one neuroactive drug at the moment of the cognitive evaluation (Yes/No).

### 2.4. Neuropsychological Assessment

A comprehensive motor-free battery of neuropsychological assessments that were considered reliable and reproducible measures of attention, concentration, memory, abstract reasoning, and problem-solving ability was given to all participants. The total time required to complete the battery was less than 90 min for all participants.

A first neuropsychological screening was done using an MMSE [29], and only individuals with a score of 22 or more underwent further evaluation. For the complete neuropsychological evaluation, a battery of commonly used neuropsychological tests was performed. All tests were administered and scored by a psychologist. The test battery included the tests shown in Figure 1.

Figure 1 shows the tests used in the protocol for this investigation. Appendix A lists the tests as well as their use and application.

### 2.5. Data Analysis

The statistical analysis was performed by using IBM SPSS for Windows, version 24.0. The normality of the neuropsychological variables was checked using tests of normality such as the Kolmogorov–Smirnov test. Depending if was normal distribution or not, different analyses were carried out:To compare the different variables between both groups, the Student’s T test (normal distribution) or the Mann–Whitney U test was used if normality was not assumed.The qualitative variables are presented in percentages and the quantitative variables with measures of centralization and dispersion (mean and standard deviation).

The association between qualitative variables of both groups was analyzed using the Chi-Square test or, if necessary, the Fisher test was used.

Next, analysis of covariance (ANCOVA) was carried out where both groups, subacute and chronic spinal cord injuries, were compared using the neuropsychological assessment protocol, as well as the scores of the state anxiety and trait anxiety scale and the depression inventory, adjusting for level of injury and psychotropic use these variables were statistically significant between the two groups compared: subacute and chronic). The significance level was set at *p* < 0.05.

## 3. Results

Characteristics of the 100 participants with SCI (50 subacute and 50 chronic) are summarized in Table 1, which shows the sociodemographic characteristics collected from the sample of one hundred SCI patients, fifty in the subacute stage and fifty in the chronic one. There were no significant differences between the two groups when age, sex, educational level, or geographical location were studied.

The mean age of the subacute SCI participants’ group was 46.82 ± 15.77 and 47.80 ± 13.75 in the chronic group. A higher proportion of males (60% approximately) vs. females was also observed.

Respective clinical characteristics and toxic habits are summarized in Table 2.

Regarding to toxic habits, 25–30% of participants were alcohol consumers when both subacute and chronic participants were evaluated. Psychotropic use was found to be in the range of 6–8% respectively, and the smoking habit was 25%.

Psychotropic use was found in 90% of participants in the subacute group, and a lower value of 74% was observed in the chronic one.

The emotional state was collected using the Beck depression inventory (BDI) and the state and trait anxiety questionnaire (STAI S/T). The results are presented in Table 3. Comparisons of mean scores obtained on the depression and anxiety scales were carried out using the BDI and STAI respectively. Adjustments were made, through regression analysis of the variable’s injury level and psychotropic use. There were no significant differences between any of the comparisons studied between the mean scores obtained, showing that the groups do not differ regarding their levels of anxiety and depression. The most important difference was the tendency to more depressive mood found in chronic SCI in comparison with the subacute group, but no significant differences.

Therefore, differences that may appear in the cognitive state would not be due to differences in their emotional state.

### Results of the Neuropsychological Assessment Protocol Analysis

Comparison of the neuropsychological assessment between subacute and chronic groups of people with SCI showed altered cognitive function (worsening with respect to the subacute group, with a medium or large effect size between groups) in the attention, processing speed, and visual memory. Data and statistics are reported in Figure 2.

With respect to other cognitive functions: the domain of memory, learning, and recognition (discriminability), showed a worsening tendency in the chronic group. Data and statistics are reported in Figure 3.

Comparison of the neuropsychological assessment between subacute and chronic SCI showed altered cognitive function with a large or very large effect in the learning and memory (RA1, RA5, RAT), executive functions (cued recall intrusions), and unrecognition (false positive) areas. Data and statistics are reported in Figure 3. The comparison between two groups proved to be worsening with respect to the subacute group.

Figure 2 shows the scores obtained in the mini-mental, digit span (WMS-III), mental control (WMS-III), and visual memory (Barcelona) tests.

Figure 3 shows the scores obtained from the TAVEC test (Test de aprendizaje verbal España–Complutense).

## 4. Discussion

The present study describes significant differences in several tests studying cognitive function in both subacute and chronic SCI participants.

More in detail, a worse performance was observed in short-term memory, long-term memory, recognition memory, free recall, and cued recall. These data show that chronic SCI participants have an initial encoding information deficit more severe than those observed in subacute SCI. In consequence, it affects the whole learning process.

Rehabilitation after SCI involves the acquisition of new skills and new knowledge. This process requires attention span, ability to concentrate, understanding, information processing, retrieval, integration, and utilization of this information. Achievement of goals in rehabilitation includes maximum independence, emotional adjustment to disability, and complete reinsertion into the community after discharge.

In this context, the big prevalence of cognitive impairment in SCI participants has an important effect in their daily living. For this reason, the purpose of this research has been to analyze and to compare the differences between several cognitive functions in two groups of spinal-cord-injured patients, both subacute and chronic. Therefore, the different scores obtained in the neuropsychological protocol designed for this purpose have been studied.

We must also consider that there is a worse performance in attentional capacity, processing speed, working memory, and visual memory. All these indexes were extrapolated from the neuropsychological tests carried out (digits and mental control in the Wechsler memory scale and visual memory of the Barcelona test) and will directly affect the performance of memory tasks and learning collected from TAVEC. This result is in line with the data obtained from the degree of learning, measured by the number of hits given in the five consecutive trials of the first part of the immediate recall test.

It has also been shown that chronic SCI participants use poorer semantic strategies to those used by subacute SCI participants. This may indicate a difficulty in establishing a prior work plan for both learning and retrieval of information, which results in a poor performance in planning the activity to be performed. If we consider also the fact that they present intrusions and perseverations more frequently, we can hypothesize that the group of chronic participants presents an inadequate capacity of inhibition and a probable disorganization of the semantic store when compared to the group of subacute ones.

Before the elaboration of this article, Molina et al., performed a similar study to this investigation with sixty-six participants with a traumatic SCI. Comparison of the neuropsychological assessment between the two groups of people with SCI showed altered cognitive function (worsening with respect to the subacute group, with a medium or large effect size between groups) in the domain of memory, learning, and recognition. The prevalence of cognitive dysfunction in the chronic stage was also confirmed at the individual level. Up to 50% of participants with chronic SCI presented more than three abnormal tests, compared to approximately 20% of participants with subacute SCI (*X*^2^ = 5.9; *p* = 0.014) [30].

Other authors. In a similar study to this research, achieved with multiple sclerosis (MS) patients, where intrusions and perseverations were used in the same sense of identification with executive functions as those used in the work of Higueras and colleagues (2009). There were no differences in the number of intrusions between the control group and the MS group. It was concluded, therefore, that the memory problem could not be related to a deficit in the executive functions, since a high number of intrusions could be reflecting a smaller capacity of inhibition of the frontal lobe [31,32].

On the other hand, regarding the discrimination index, statistically significant differences were observed between both groups, the group of chronic spinal-cord-injured patients obtained lower scores, as they reached higher scores in both the number of false positives and omissions, according to Molina et al. (2018) [30].

The optimal performance in episodic memory tests of the word list, such as those used in this work, depends on whether processes such as coding, storage or consolidation, and retrieval are intact. According to Chertkow et al. (2007), a poor performance in free recall and cued recall would be indicative of a problem in coding, and this pattern is followed by chronic SCI participants if compared to subacute SCI participants [33].

In conclusion, the present study shows differences in the learning capacity, memory types, encoding indexes, recovery indexes, and discrimination indexes between both groups. Concretely, chronic SCI obtained significant lower values that obtained by the subacute SCI participants. Considering that different prefrontal regions are activated during the encoding and recovery processes (over the left hemisphere for encoding and right hemisphere for recovery), it could be inferred, in a very careful way, that it is possible that an alteration of these structures could explain the differences observed between the studied groups.

The possible alteration could be caused by the trauma itself [34,35,36] or due to complications raised in the subacute process and/or during hospitalization [37,38].

Bearing in mind the results obtained, we demonstrate that chronic SCI participants, have a mnesic and learning performance significantly lower than the subacute SCI participants.

## 5. Conclusions

We therefore can conclude that:Chronic SCI participants obtained lower scores in the neuropsychological assessment protocol than those obtained by subacute injured participants. Most of them obtained significant scores.These differences are not attributable to differences between groups in terms of sex, age, level of education, region of origin, and mood (anxiety and depression). There were also no differences in the presence of toxic habits.Psychotropic use was a more prominent factor in the group of subacute participants compared to the group of chronic participants.

We can conclude that chronic SCI participants presented significantly lower scores in the neuropsychological assessment protocol than the subacute SCI participants’ group, and a clear attributable factor couldn’t be determinate, except for the years from injury.

## 6. Limitations of the Study

The analysis of the present study has been conditioned by the limitations and difficulties of this type of research with participants suffering spinal cord injury.

One limitation of the present study was the small size of the studied cohort and its heterogeneity. However, it should be considered that participants were recruited during the hospitalization stage, considering relevant aspects such as age, sex, etiology, injury level, and other sociodemographic factors.

Another important limitation was the lack of data about the cognitive status prior to the spinal cord injury to disclose a change in cognition induced by the SCI.

Further research based on other techniques such as neuroimaging and neurophysiological exploration, could offer more information about the brain function related to these findings.

Finally, it should be noted that currently there are a few studies that have used TAVEC to study cognitive performance in participants with SCI in any of its stages. So, we cannot faithfully compare the results obtained in this research. There are studies applying other batteries of tests and developed in different stages of the injury. Most of them compare their data with a control group that in many cases is composed of subjects without spinal cord injury. In addition, there is evidence that the existing studies on this subject are composed in the great majority of cases of population with spinal cord injury and traumatic brain injury.

With this study we would like to highlight that an early cognitive intervention in the spinal cord injury would improve functional performance in people at risk of suffering from cognitive impairment. Designing effective prevention programs from the first moments of spinal cord injury and that their planning and evaluation are a constant throughout the rehabilitation period.

All this would have as a consequence that rehabilitation periods would be more optimal; reducing morbidity, mortality, and hospital stay times; and optimizing preventive resources.

## Figures and Tables

**Figure 1 healthcare-09-00241-f001:**
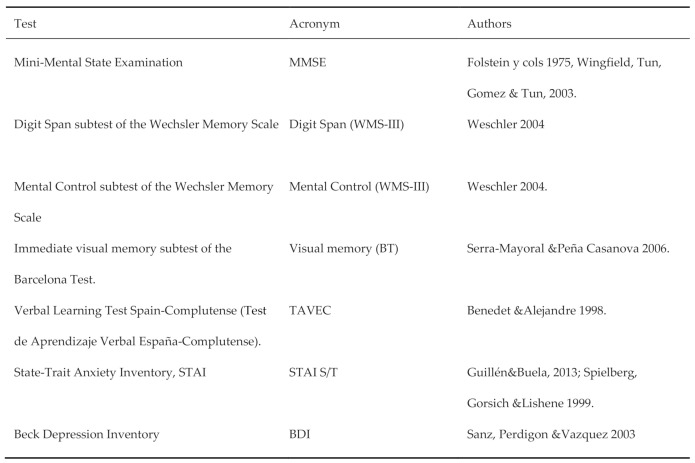
Tests used in the neuropsychological assessment protocol in patients with spinal cord injury. Source: Own elaboration of the authors.

**Figure 2 healthcare-09-00241-f002:**
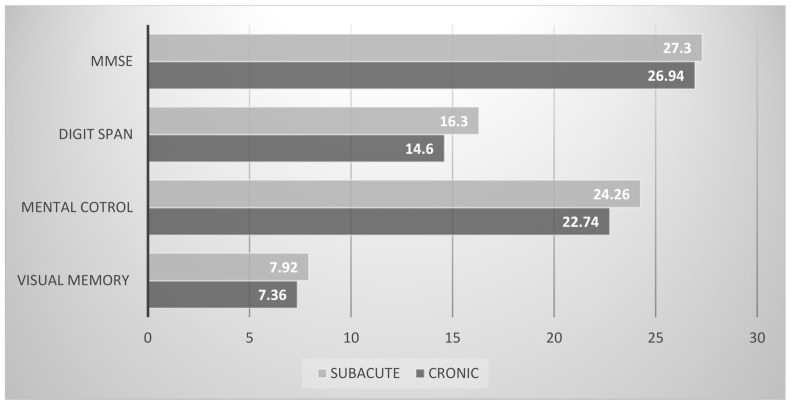
MMSE, digit span (WMS-III), mental control (WMS-III), and visual memory (Barcelona test) scores in SCI patients. Legend: MMSE: mini-mental State Examination; digit span (WMS-III): digit span subtest of the Wechsler memory scale; mental control (WMS-III): mental control subtest of the Wechsler memory scale; BT: Barcelona test.

**Figure 3 healthcare-09-00241-f003:**
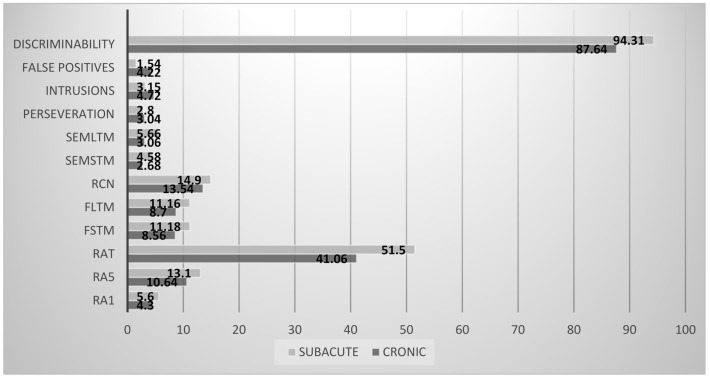
Test de aprendizaje verbal España–Complutense (TAVEC) scores in SCI patients. Legend: RA_1_: hits in test 1 of the A list; RA_5_: hits in test 5 of the A list; RA_T_: sum of the hits in the five trials; FSTM: free short-term memory; FLTM: free long-term memory; RCN: recognition; SemSTM: semantic short-term memory; SemLTM: semantic long-term memory; FP: false positives.

**Table 1 healthcare-09-00241-t001:** Sociodemographic characteristics of spinal cord injury (SCI) patients.

Sociodemographic Characteristics	Categories	Subacute *N* = 50 *N* (%)	Chronic *N* = 50 *N* (%)	Statistics
Age		46.2 (15.77)	47.80 (13.75)	t = 0.331 *p* = 0.74
Sex	Male Female	30 (60%) 20 (40%)	31 (62%) 19 (38%)	*X*^2^ = 0.42 *p* = 0.83
Level of study	Reading/Writing Basic. Technical school. High school University.	3(6%) 21 (42%) 9 (18%) 10(20%) 7 (14%)	5 (10%) 25 (50%) 7 (14%) 5 (10%) 8 (16%)	*X*^2^ = 10.33 *p* = 0.06
Region of origin	Abroad. Madrid. Castilla la Mancha. Castilla León. Andalucía. Resto de España.	7 (14%) 9 (18%) 6 (12%) 9 (18%) 9 (18%) 10 (20%)	1 (2%) 2 (4%) 4 (8%) 16 (32%) 4 (8%) 22 (44%)	*X*^2^ = 25.26 *p* = 0.09

N: Number of patients; %: percentage; t: Student’s T; *X*^2^: Chi-Square; *p*: significance.

**Table 2 healthcare-09-00241-t002:** Clinical characteristics and toxic habits of SCI patients.

Clinical Characteristics and Toxic Habits	Categories	Subacute *N* = 50 *N* (%)	Chronic *N* = 50 *N* (%)	Stadistics *X*^2^/*p*
Injury Level	Cervical	22 (44%)	15 (30%)	*X*^2^ = 0.3.14
	Dorsal	23 (46%)	28 (56%)	*p* = 0.369
	Lumbar	3 (6%)	6 (12%)	
	Sacrum–coccyx	2 (4%)	1 (2%)	
ASIA	A	12 (24%)	25 (50.6%)	*X*^2^ = 8.23
	B	7 (14%)	7 (14%)	*P* = 0.041
	C	16 (32%)	8 (16%)	
	D	15 (30%)	10 (20%	
Causes of Injury	Traumatic	32 (64%)	34 (68%)	X*^2^* = 0.17
	Non-traumatic	18 (36%)	16 (32%)	*p* = 0.0673
Traumatic	Traffic accidents	12 (37.5%)	19 (55.8%)	*X*^2^ = 13.79
	Casual fall	7 (21.8%)	5(14.7%)	*p* = 0.314
	Precipitation	2 (6.2%)	-	
	Dives	2 (6.2%)	-	
	Work accidents	4 (12.5%)	4 (11.7%)	
	Others	3 (15.6%)	12 (35.2%)	
Non-Traumatic	Myelitis	5 (27.7%)	3 (18.7%)	*X*^2^ = 4.54
	Spinal infarction	2 (11.1%)	2 (12.5%)	*p* = 0.474
	Surgical intervention	7 (38.8%)	8 (50%)	
	Arteriovenous malformation	1 (5.5%)	1 (6.2%)	
	Others	3(16.6%)	2 (12.5%)	
Alcohol Consumption	Yes	14 (28%)	12 (24%)	*X*^2^ = 0.877
	No	36 (72%)	40 (80%)	*p* = 0.349
Drugs	Yes	4 (8%)	3 (6%)	*X*^2^ = 0.154
	No	46 (92%)	47(94%)	*p* = 0.695
Psychotropic use	Yes	45 (90%)	37 (74%)	*X*^2^ = 4.36
	No	5 (10%)	13 (26%)	*p* = 0.037

ASIA: American Spinal Injury Association Impairment Scale; N = Number of patients; % percentage; *X*^2^ Chi-Square; *p*: significance.

**Table 3 healthcare-09-00241-t003:** Regression analysis of the anxiety scores (state-trait) and depression, adjusted by level of injury and psychotropic use.

Test	Group	*N*	M (SD)	CI (95%)	*p^c^*	*p^a^*
BDI	Subacute	50	7.82 (6.73)	(5.91–9.73)	*p^c^* = 0.62	*p^a^* = 0.52
	Chronic	50	8.52 (7.45)	(6.40–10.64)		
STAI-S	Subacute	50	20.04 (10.18)	(1.15–22.93)	*p^c^* = 0.55	*p^a^* = 0.44
	Chronic	50	21.4 (12.79)	(17.76–25.04)		
STAI-T	Subacute	50	21.04 (12.05)	(17.62–24.46)	*p^c^* = 0.35	*p^a^* = 0.47
	Chronic	50	18.7 (13.25)	(14.93–22.47)		

Legend: BDI: Beck depression inventory; STAI-S: anxiety state; STAI-T: anxiety trait; CI: confidence interval; *p^c^*: level of significance; *p^a^*: level of significance adjusted for level of injury and psychotropic use.

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
