# Peer review of "Neuropsychological Study in Patients with Spinal Cord Injuries"

_healthcare, 2021, doi:10.3390/healthcare9030241_

Round 1

Reviewer 1 Report

The paper is about a very interesting topic but it is necessary to improve its quality. 

The discussion is very similar to the Results. I suggest to better describe results taking into account previous papers. I also suggest to increase the quality of description and to insert some more bibliographic references. Sometimes the authors quote the findings of an article but they do not insert reference. 

I also suggest to better describe the relationship between cognitive impairments and difficulties and depression, as People with Spinal Cord Injury have an high prevalence of depression. 

Last but not least, I suggest to modify the title: there is no reference to Spinal  Cord Injury, only reference to injury but the article is on Spinal cord injury. 

Author Response

I have made the changes you have proposed. I have changed the title. 

The results have also been modified, describing results taking into account other papers. I have clarified the prevalence of depression in these patients with spinal cord injuries. 

The variables depression and anxiety have been measured in both groups and no significant differences were found. 

I have reviewed the quote and I insert it reference

I hope you like it and thank you very much for your contributions. 

Reviewer 2 Report

This study compares the differences in cognitive deficits of people with subacute vs chronic spinal cord injury. To do this, the authors have used a set of different neuropsychological protocols that allowed them to assess the cognitive status of the subjects. The authors have carried out anxiety tests to verify that the results obtained are not caused by differences in anxiety between the 2 groups studied, this reinforces the results shown, however, I mention certain aspects that, in my opinion, should be revised to improve the manuscript:

In line 42 an abbreviation "MV" is written but it is not indicated what it means.

In section “1” of the objective of the study, not only the objective is mentioned but also refers to what has been done “The scores obtained from both groups were analyzed and compared”, which could be understood as part of the objective "2".

In the conclusions "lines: 331-334" the authors say "Authors should discuss the results and how they can be interpreted from the perspective of previous studies and of the working hypotheses. The findings and their implications should be discussed in the broadest context possible. Future research directions may also be highlighted. " This is not a conclusion, it is a comment that highlights something that is missing in the manuscript: a better discussion of the implications of the results and explain how this study could be the basis for future research. It is also missing a comparison between the results obtained with previously existing information but this point is justified by the authors in the section “Limitations of the Study”.

The section “Limitations of the Study” is a noteworthy and positive point because it shows that the authors are aware of the limitations, a fact that is also noted in the caution they show when writing the conclusions.

My biggest concern is about statistics. In the "Materials and Methods" it is detailed how the statistical analysis has been carried out, however, the statistical data is missed, for example, in the text, when referring to figures 2 or 3 the authors says: "data and statistics are reported in Figure x”, however, in the figures I can only see the scores obtained but not the statistical data.

This work aims to provide new and interesting information on cognitive deficits in people with spinal cord injury that could contribute to the development of therapies that promote the recovery, however, in my opinion, it is necessary to improve the manuscript a little before being published.

Author Response

Hello, Thank you for your contributions. I have made the changes you have proposed. 

I have clarified line 43 it was an error. I have reviewed and clarified the objetives. 

In conclusions, I have withdrawn the paragraph that you suggest me. 

Above all I have tried to improve the statistical analysis part. 

I hope you like it and thank you very much.

I send you the new manuscript. 

Round 2

Reviewer 1 Report

The authors addressed all the issues .

Reviewer 2 Report

The authors have made a revision of the manuscript that has improved it significantly. The improvement is from the change in the title to the conclusions. Especially the statistics explanation, they have written it in a clearer way.

This work provides new interesting information on cognitive deficits in people with spinal cord injury that could contribute to the development of therapies that promote the recovery. In my opinion, the manuscript is now of sufficient quality to be published.